

# MILQ: A liquefaction triggering map for Italy

Simone Barani[1], Gabriele Ferretti[1], Davide Scafidi[1]

[1]Dipartimento di Scienze della Terra dell'Ambiente e della Vita, Università di Genova, Genova, 16132, Italy

*Correspondence to*: Simone Barani (simone.barani@unige.it)

**Abstract.** In the present study, we analyze ground-motion hazard maps and hazard disaggregation in order to define areas in Italy where liquefaction triggering due to seismic activity can not be excluded. To this end, we refer to the triggering criteria proposed by the Italian Guidelines for Seismic Microzonation, which are described in the main body of the manuscript. However, the study can be replicated in other countries that adopt different criteria. The final goal is the definition of a screening map for all of Italy that classifies sites in terms of liquefaction triggering potential according to their seismic

hazard level. The map, which is referred with the Italian acronym MILQ (Mappa del potenziale d'Innesco della LiQuefazione), and the associated data are freely accessible at the following web address: www.distav.unige.it/rsni/milq.php. Our results can be useful to guide land-use planners in deciding whether liquefaction is a hazard that needs to be considered within the planning processes or not. Furthermore, they can serve as a guide for recommending geological and geotechnical investigations aimed at the evaluation of liquefaction hazards or, conversely, rule out further studies with a consequent

saving in efforts and money.

## 1 Introduction

The assessment of potential liquefaction hazards always involves two questions: (1) are the geological conditions at the site of interest favorable to liquefaction? (2) will liquefaction be triggered by future earthquake ground motions at that site? The latter is usually asked in case of a positive answer to the first question, namely: if the site is susceptible to soil liquefaction,

will liquefaction be triggered by future seismicity? While answering the first question may involve costly in-situ investigations mainly aimed at defining the depth of the water table and soil granulometry, the answer to the second question can be simply obtained by querying national hazard maps via online web services, at least at the screening level. The scope is to get the value of the two ground-motion parameters commonly used in liquefaction hazard assessments: peak horizontal acceleration (in the present work, the terms "peak horizontal acceleration" and "peak ground acceleration", as well as the

relative acronyms PHA and PGA, are used interchangeably) at the ground surface (or the return period of interest and earthquake magnitude (e.g., maximum magnitude, mean or modal magnitude from hazard disaggregation). These parameters are related to the duration and intensity of ground shaking and are key factors of the well-known simplified approach of Seed and Idriss (1971) for evaluating liquefaction resistance of soils (see also Youd et al., 2001). Thus, it may be convenient to reverse the question above: if liquefaction can be triggered by seismic activity at the site of interest, is the site susceptible to



soil liquefaction? Indeed, if the site hazard is deemed too low to trigger liquefaction (i.e., if the values of the surface PGA and earthquake magnitude for that site are below some threshold values), then one can avoid recommending specific investigations aimed at the geological and geotechnical characterization of the subsoil. This reversed decision scheme can be useful to guide land-use planners in deciding whether liquefaction is a hazard that needs to be considered within the planning processes or not.

With the increased availability of data in recent years, substantial research has been carried out to establish thresholds of magnitude and PGA at the ground surface below which the possibility of triggering liquefaction can be discounted. Although the use of a minimum threshold magnitude is controversial (Musson, 2020; Green and Bommer, 2019), available studies converge to a value of 5.0 (e.g., Atkinson et al., 1984; Green and Bommer, 2019), while a threshold of about 0.1g has been suggested for the PGA at the ground surface (e.g., Santucci de Magistris et al., 2013; Santucci de Magistris, 2015). However,

exceptional liquefaction phenomena induced by minor events (i.e., with magnitude less than 5.0) have been observed in extremely susceptible soils (e.g., Martino et al., 2014; Santucci de Magistris (2015) and references therein; Zimmaro et al., 2019; Brandenberg et al., 2020; Bozzoni et al., 2021). Such values are adopted by the Italian building code (Ministero delle Infrastrutture e dei Trasporti, 2018[1]) as well as by the seismic microzonation guidelines for Italy (SM Working Group, 2008 and 2015).

The scope of the present work is to define a liquefaction triggering map of Italy that can be used for seismic microzonation purposes and, therefore, can guide land-use planning within the framework of risk mitigation programs. To this end, we combine data from the reference Italian seismic hazard maps (MPS Working Group, 2004; Stucchi et al., 2011) and hazard disaggregation (Barani et al., 2009), which is here expressed in terms of 1D magnitude contributions to the hazard. Note that the results presented in this work do not address the issue of liquefaction assessments for structural design, for which the

hazard corresponding to multiple return periods should be taken into account. In this work, we focus on the 475-year return period according to the Italian Guidelines for Seismic Microzonation (hereinafter, we will use the ICMS acronym, commonly used in Italy). The reliability of the results is checked by comparing them with observations of past liquefaction events in Italy (Martino et al., 2014; Caprari et al., 2018), and in the light of recent case studies associated with the 2012 Emilia seismic sequence (e.g., Minarelli et al. 2022).

Although we are aware that liquefaction hazard requires site-specific geological and geotechnical investigations (liquefaction is indeed a highly localized phenomenon), our study may serve as reference guide for identifying sites where the possibility of triggering liquefaction can be discounted. Moreover, it provides basic ground-motion data for evaluating liquefaction resistance of soils at specific sites and for defining liquefaction susceptibility maps (e.g., Zumpano et al., 2022). Such data are freely available through the web service at www.distav.unige.it/rsni/milq.php.

---

[1]It is worth noting that, compared to the original version released in 2008 (Ministero delle infrastrutture e dei Trasporti, 2008), the 2018 update of the Italian building code does not specify a lower bound magnitude for liquefaction triggering.





## 2 Criteria for liquefaction triggering

According to the ICMS, liquefaction is expected to occur if the following conditions are met (SM Working Group, 2008 and 2015):

1. the lithological sequence presents layers of non-cohesive, saturated soil (sandy limes, sands, limey sands, gravely sands, clayey sands and sandy gravels) down to 20m depth
2. the average depth of the water table is within 15m of the ground surface
3. expected seismic events must be characterized by magnitude values $M_w \geq 5$ and must produce a surface peak ground acceleration $a_{max} \geq 0.1$g.

If even one of the conditions above is not verified, we can assume that the area under study is not susceptible to liquefaction. In this study, our focus is on the third condition. Contrary to the first two ones, it does not require local data, at least at a screening level (corresponding to the so-called Level 1 Microzonation in the ICMS). We use national hazard maps along with the corresponding hazard disaggregation results to define those sites in Italy where this condition is not met – that is, where the triggering of liquefaction phenomena is unlikely and can be excluded. If the condition is not met, then we can avoid recommending specific geotechnical investigations aimed at verifying the first two criteria, with a consequent saving of efforts and money.

## 3 Methodology

In order to define those areas in Italy where seismic activity could trigger liquefaction (according to the third condition listed in the previous section), we analyze the reference seismic hazard data for the national territory. First, the PGA hazard for rock conditions and flat topography associated with a return period of 475 years (MPS Working Group, 2004; Stucchi et al., 2011) is modified to incorporate site amplification due to local geology and irregular topography (recall that the 475-year return period is assumed as the single, reference return period by the ICMS). Essentially, the resulting $a_{max}$ map allows us to define areas with high (i.e., $a_{max} \geq 0.1$g) and low (i.e., $a_{max} < 0.1$g) local seismic hazard. Then, 1D disaggregation results (i.e., magnitude disaggregation) are produced based on the reference work of Barani et al. (2009) and analyzed to define those sites where the hazard is controlled by scenarios with magnitude $M_w \geq 5$. As sites respond at specific characteristic frequencies (depending on local geological characteristics) and disaggregation results may vary significantly with response period (*T*), disaggregation analysis is carried out for different values of *T*. Particular attention is paid to the choice of mean or modal magnitude values as reference scenarios, in order to avoid both non-conservative and over-conservative scenarios. A cross analysis of hazard and disaggregation data is finally performed via the QGIS software (https://www.qgis.org/) to define the sites where liquefaction triggering is expected to occur.



### 3.1 Seismic hazard map

In order to define those sites with surface peak horizontal acceleration $a_{max} \geq 0.1g$ for a return period of 475 years, we refer to the Italian seismic hazard map adopted by the national building code to define the design seismic action (MPS Working Group, 2004; Stucchi et al., 2011). We recall here that this map was developed assuming flat topography and rock conditions. Hence, following the definition of surface peak horizontal acceleration, where the word "surface" is used to indicate that $a_{max}$ refer to site-specific conditions, we first need to amend the reference hazard in order to incorporate possible ground-motion amplification effects related to sub-surface lithological conditions and topographic irregularities. According to the updated Italian building code (Ministero delle Infrastrutture e dei Trasporti, 2018), the rock seismic action at a given site is corrected by a site term ($S$) that accounts for both soil and topographic conditions (i.e., $S = S_S \times S_T$ where $S_S$ and $S_T$ are the soil and topographic amplification factors, respectively). For that site, the values of $S_S$ and $S_T$ can be defined according to simple site classification criteria, which are not reported here for the sake of brevity. While the soil classification is mainly based on the values of the $V_{S,eq}$ parameter[2] (i.e., time-averaged shear-wave velocity above the seismic bedrock, the latter being defined as the rock formation or rigid soil with $V_S \geq 800m/s$), the topographic classification is based on site acclivity. Recently, Forte et al. (2019) developed a soil classification map for all of Italy in compliance with the ground types defined by the updated Italian building code[3]. Mascandola et al. (2021a) did the same for topography, providing a topographic classification map of Italy. We refer to these works to amend the reference PGA hazard map for rock conditions and flat topography associated with a return period of 475 years (i.e., $a_g$ hazard map). In compliance with the Italian building code, $a_{max} = a_g \times S$. The maps in Fig. 1 show the geographic distributions of $S_S$ and $S_T$ using the same grid of Forte et al. (2019), which is adopted as reference to produce the final results. The comparison between the original $a_g$ hazard map for a return period of 475 years and the corresponding $a_{max}$ map is presented in Fig. 2. As expected, incorporating the site term into the hazard increases the number of sites where liquefaction is likely to occur. The number of sites with $a_{max} \geq 0.1g$ grows from 753820 to 883363.

### 3.2 Hazard disaggregation

In order to define the sites where the seismic hazard is controlled by $M_w \geq 5$ scenarios, we have disaggregated the hazard corresponding to a return period of 475 years for all computation nodes considered in the hazard assessment of Italy (MPS Working Group, 2004; Stucchi et al., 2011). Specifically, the contributions from different magnitude scenarios are determined (by summation) from the joint $M$-$R$-$\varepsilon$ distributions originally computed by Barani et al. (2009), where $R$ is the

---

[2]Note that, in Italy, $V_{S,eq}$ has replaced the well-known $V_{S,30}$ parameter, which assumes a fixed depth of 30m for the calculation of the time-averaged shear-wave velocity in the subsoil.

[3]For the sake of completeness, Forte et al. (2019) also developed a similar map based on the $V_{S,30}$ parameter, parameter which was adopted by the elder Italian norms (Ministero delle Infrastrutture e dei Trasporti, 2008) and Eurocode 8 (European Committee for Standardization, 2004) for the purpose of site classification. Almost concurrently, a $V_{S,30}$ map was developed by Mori et al. (2020).





source-to-site distance and $\varepsilon$ indicates the ground-motion error term. That work still represents the reference disaggregation study for Italy and, as such, is taken as reference by the ICMS. Accordingly, we consider bins of 0.5 magnitude units. Note that, according to Iervolino (2016), there is no need to consider the site correction term $S$ in the disaggregation process. In our application, indeed, disaggregation results are invariant with soil category.

As the ICMS does not specify how to handle response periods in the disaggregation process (i.e., it is not specified if the controlling magnitude should result from the disaggregation of the hazard for a specific value of $T$), we have disaggregated the spectral acceleration hazard associated with periods of 0.01s (i.e., PGA), 0.2s, and 1s. While PGA-hazard disaggregation may be appropriate for rock sites (which are known to resonate at high frequencies, and where one can exclude the possibility of liquefaction), it can provide non-conservative results for most soil sites, which generally have resonance periods in the 0.1 – 1s range (this range may extend up to longer periods in the case of deep alluvial valleys). It is known, indeed, that the contribution from larger magnitudes increases as $T$ increases. Therefore, disaggregating the hazard for different spectral periods allows us to define the controlling magnitude in relation to geological conditions (through site classification). We expect that decreasing $V_{S,eq}$ and/or increasing soil thickness the site resonance period increases. Therefore, for those sites characterized by lower values of $V_{S,eq}$ and/or thicker soils (i.e., site classified as ground type C, D, or E in the Italian building code), it seems reasonable to define the controlling earthquake based on the disaggregation of the hazard at longer periods.

The maps in Fig. 3 show the geographic distribution of mean and modal magnitudes ($\bar{M}$ and $M^*$, respectively) for the three periods considered. Comparing the maps for the same period immediately reveals significant differences between mean and modal scenarios. On average, larger magnitudes control di hazard in areas characterized by either a higher seismic activity (e.g., Central and Southern Apennines) or very low seismicity (e.g., Southeastern edge of the Italian peninsula where the hazard is controlled by large magnitude, distant events), while lower magnitudes dominate in areas of mild-to-moderate but relatively frequent seismic activity (e.g., some areas in the Po Plain and western Alps). However, with the same area, the values of $\bar{M}$ and $M^*$ may vary substantially. We observe that, the maps of $\bar{M}$ are generally more conservative than those of $M^*$, especially in areas where the hazard is lower (e.g., northern Italy). Which earthquake scenario should be considered as representative of the site hazard? The mean or the mode? Moreover, which mode should be considered in the case of bi-modal distributions? These questions are still open in the scientific community, in the sense that there is no common opinion about the choice of the mean or the mode. It is well known that the mode has the clear advantage of representing the most likely scenario but is sensitive to the binning scheme adopted. On the other hand, the mean is not sensitive to the bin size, but it might not represent the most likely scenario or, in some cases (not so infrequent, particularly in the case of bi-modal distributions), it could represent an unlikely one. In this work, we solve the crucial issue of the controlling earthquake by adopting the conservative view that stronger is safer. Specifically, for each period considered, we assume the following:



- in the case of unimodal distributions with no skew (i.e., symmetric or nearly symmetric about the mean) (Fig. 4a), either the mean or modal magnitude can be indiscriminately taken as the preferred magnitude ($\widehat{M}(T) = \bar{M}(T) = M^*(T)$)

- in the case of unimodal, negatively skewed distributions (Fig. 4b), the modal magnitude is assumed as the preferred magnitude ($\widehat{M}(T) = M^*(T)$)

- in the case of unimodal, positively skewed distributions (Fig. 4c), the mean magnitude is assumed as the preferred magnitude ($\widehat{M}(T) = \bar{M}(T)$)

- in the case of bimodal distributions, the second mode ($M_2^*$) is taken as the preferred magnitude ($\widehat{M}(T) = M^*(T)$) if its contribution to the hazard is greater than the contribution associated with the mean magnitude (Fig. 4d). Conversely, we assume that its contribution to the hazard is negligible, and the mean, which reflect that contribution to some extent, is assumed as the preferred magnitude ($\widehat{M}(T) = \bar{M}(T)$) (Fig. 4e). Thus, this latter case resembles the case of a unimodal, positively skewed distribution (see the previous bullet point) and avoids selecting over-conservative scenarios with very small contributions (i.e., unlikely scenarios).

The maps resulting from the application of the criteria above are shown in the left column of Fig. 5 for the three spectral periods considered. For each of them, the maps in the right column show the geographic distribution of the contributions associated with $\widehat{M}(T)$. It can be observed that such contributions are generally larger than 10%. As expected, the maps in the left column show that the values of $\widehat{M}$ increase with increasing spectral period, reaching the largest values in Southern Italy for $T = 1.0$s where $\widehat{M}$ is between 7.0 and 7.5. Note the similarity of the maps corresponding to $T = 0.01$s (i.e., PGA) and $T = 0.2$s, with the latter being slightly more conservative. Despite this similarity, we prefer to consider both these maps in order to distinguish between rock sites, for which PGA-hazard disaggregation is geologically consistent, and sites characterized by deposits of dense soil, which are expected to resonate at longer periods.

## 4 Results

The three maps shown in Fig. 5 are used in conjunction with the map of site classification (Fig. 9b in Forte et al., 2019) to define the final map of $\widehat{M}$ (from here on, we drop the dependence of $\widehat{M}$ on $T$ to indicate the preferred magnitude) according to the following criteria:

- $\widehat{M} = \widehat{M}(T = 0.01$s$)$ for ground type A sites (i.e., rock sites or stiff soils with $V_S \geq 800$m/s)

- $\widehat{M} = \widehat{M}(T = 0.2$s$)$ for ground type B sites (i.e., soft rock or deposits of dense soil characterized by a gradual increase of the mechanical properties with depth, and $360 \leq V_{S,eq} < 800$m/s)

- $\widehat{M} = \widehat{M}(T = 1.0$s$)$ for ground type C, D, and E sites (i.e., sites characterized by deposits of loose-to-medium cohesionless soil with thickness either greater (C and D sites) or smaller than 30m (E sites), and $100 \leq V_{S,eq} < 360$m/s)



The resulting map of $\hat{M}$ is shown in Fig. 6.

Combining the maps in Fig. 6 and Fig. 2b leads to the liquefaction triggering map shown in Fig. 7, which is referred with the Italian acronym MILQ (Mappa del potenziale d'Innesco della LiQuefazione). The color scale has been chosen to divide sites into classes of increasing liquefaction triggering potential (LTP), from no potential (i.e., $a_{max} < 0.1$g and $\hat{M} < 5.0$) to very high potential (i.e., $a_{max} \geq 0.2$g and $\hat{M} \geq 6.0$). Analyzing the map, we observe that:

- 9.8% of nodes fall in areas with no liquefaction triggering potential (class LTP-0: $a_{max} < 0.1$g and $\hat{M} < 5.0$); they
are mainly concentrated in very few areas in the northwest and in almost all of Sardinia Island

- 20.4% of nodes have very low liquefaction triggering potential (class LTP-1: $a_{max} < 0.1$g and $5.0 \leq \hat{M} < 6.0$ or $0.1$g $\leq a_{max} < 0.2$g and $\hat{M} < 5.0$); most of them are concentrated in areas characterized by low-to-moderate ground-motion hazard, controlled by small-to-moderate magnitude events (e.g., northern Italy, areas along the northern Tyrrhenian coast)

- 8.8% of nodes have low liquefaction triggering potential (class LTP-2: $a_{max} < 0.1$g and $\hat{M} \geq 6.0$ or $a_{max} \geq 0.2$g and $\hat{M} < 5.0$); most these nodes are in low seismicity areas (e.g., northern-eastern Alps and south-eastern Italy) where the hazard tends to be controlled by stronger, generally distant, events or, conversely, in areas of higher hazard but controlled by low magnitudes (e.g., northern Sicily)

- 10.5% of nodes have moderate liquefaction triggering potential (class LTP-3: $0.1$g $\leq a_{max} < 0.2$g and $5.0 \leq \hat{M} <$
$6.0$); most of them are concentrated in northern Italy, especially in the western sector and in the Po Plain, and in central Sicily, where both the ground-motion hazard and $\hat{M}$ are moderate

- 31.6% of nodes have high liquefaction triggering potential (class LTP-4: $0.1$g $\leq a_{max} < 0.2$g and $\hat{M} \geq 6.0$ or $a_{max} \geq 0.2$g and $5.0 \leq \hat{M} < 6.0$). Most of them are concentrated in areas of increased seismic activity in central and southern Italy, and in the northeast; the nodes with the highest values of $\hat{M}$ are located in areas characterized by
lower ground-motion hazard (e.g., some areas along the southern Tyrrhenian and Ionian coasts, and in central-eastern Sicily)

- 18.9% of nodes have very high liquefaction triggering potential (class LTP-5: $a_{max} \geq 0.2$g and $\hat{M} \geq 6.0$); most these nodes are concentrated in central and southern Italy, following the Apennine arc down to the volcanic area in eastern Sicily, and in the northeast, where the ground-motion hazard reaches the highest levels and is dominated by
(local) moderate-to-high magnitude seismicity.

## 5 Online application for data retrieval

To make our results available to land-use planners and practitioners, we have developed a web service, freely accessible at www.distav.unige.it/rsni/milq.php. The web service data are stored in a "PostgreSQL" database. All the online components



were developed in "PHP" and "HTML5" languages, to ensure adherence to current web standards. Online maps are based on
"Leaflet", an open-source JavaScript library for mobile-friendly interactive maps.

For a specified location (defined by a pair of geographic coordinates), the web service provides the values of $a_{max}$ and $\widehat{M}$
computed according to the site classifications adopted in the present study. Specifically, the values associated with the
nearest node are returned (no interpolation is performed). In addition, as the actual ground type and topographic class at a
site (e.g., resulting from site-specific data) can differ from those considered here, the service allows the user to change them
through a user-friendly interface and returns the updated values of $a_{max}$ and $\widehat{M}$ as output.

## 6 Discussion

Given the practical importance of liquefaction triggering potential maps, the latter should be subjected to testing before
application. In the present study, we examine the reliability of the map in Fig. 7 by analyzing the geographic distribution of
past liquefaction phenomena reported in the Italian Catalogue of Earthquake-Induced Ground Failures – CEDIT (Caprari et
al., 2018). These events have been superimposed on that map, which is now displayed in Fig. 8 using a more intuitive red
and white scale, with red dots indicating the sites where liquefaction triggering is expected to occur (see the caption for
further details). As is evident, most observations fall in red areas, thus indicating the consistency of our results. The matched
observations are 314 out of 328. A more quantitative analysis of the results is provided by the histogram in Fig. 9, which
shows the percentage of CEDIT observations for each class of liquefaction triggering potential defined above (from LTP-0
to LTP-5). Among the observations that fall in the red areas of the map in Fig. 8, 92% of them correspond to nodes with high
or very high liquefaction triggering potential (classes LTP-4 and LTP-5), while 4% have a moderate potential (class LTP-3).
Only 4% of observations correspond to nodes with very low or low triggering potential (class LTP-1 and LTP-2). No
observations fall in regions with no triggering potential (class LTP-0).

Despite the good agreement between our map and past observations, a few questions need to be answered. To what extent
does this matching derive from the criterion adopted to define $\widehat{M}$ in relation the soil resonance? In other words, what is the
sensitivity of our results to the choice of $\widehat{M}(T)$ for different ground types? Can one obtain similar conservative results just
assuming the disaggregation results for a single spectral period (e.g., $T = 0.01$s in line with the definition of $a_{max}$) and
increasing the return time (e.g., to 2475 years)? To answer these questions, we focus on the Po Plain area, which experienced
liquefaction in several locations during the 2012 Emilia seismic sequence (Scognamiglio et al., 2012; Minarelli et al., 2022).
The Quaternary deposits, consisting mainly of an alternation of sands, silts and clays, are rather homogeneous throughout
this area. Indeed, according to the soil classification of Forte et al. (2019) (see Fig. 9b therein), most sites in the Po Plain
basin can be classified as ground type C or D (i.e., sites characterized by deposits of loose-to-medium cohesionless soil with
thickness greater than 30m, and $100 \leq V_{S,eq} < 360$m/s). However, deep stratigraphic discontinuities can be identified (e.g.,
Mascandola et al., 2019). Such discontinuities are responsible for significant ground-motion amplification at long periods (>
1s) (e.g., Luzi et al., 2013; Abraham et al., 2015; Mascandola et al., 2021b).





For the Po Plain area, Fig. 10 compares alternative liquefaction triggering scenarios by adopting different assumptions: precisely, Fig. 10a is just a zoom of Fig. 8; Fig. 10b is the same as Fig. 10a but is obtained by assuming $\widehat{M} = \widehat{M}(T = 0.01s)$ regardless of the ground type; finally, Fig. 10c is the same as Fig. 10b but for a return period of 2475 years. As expected, comparing the map in Fig. 10b with the reference one in Fig. 10a reveals that disaggregating only the PGA-hazard can lead

to an underestimation of the liquefaction triggering potential at soil sites. Such an underestimation can be significant at sites characterized by thick and soft soil deposits, as they generally have resonance periods much larger than 0.1s. This justifies the use of the disaggregation of the hazard associated with either short or long spectral periods depending on site response. Especially for soft soil sites or sites characterized by thick soil deposits, the use of hazard disaggregation corresponding to longer periods prevents the adoption of non-conservative provisions by land-use planners or practitioners. As for the map in

Fig. 10c, which refer to a return period of 2475 years, it still provides a conservative scenario like the reference map in Fig. 10a, but that scenario is justified neither by the local geology (see above) nor by the return period of a $M_w = 5.9$ earthquake, such as the main shock of the 2012 Emilia seismic sequence with epicenter near the town of Ferrara. According to the MPS Working Group (2004), indeed, the return period of a $M_w = 5.9$ earthquake in that area is about 360 years.

**Conclusions**

Besides ground-motion amplification effects, which are undoubtedly the most important of all seismic (site) effects because of their impact on the environment and society, instability phenomena induced by earthquake ground shaking (i.e., surface faulting, ground failure, and soil liquefaction) also play a key role in defining risk mitigation strategies and land-use planning. In recent years, the identification of unstable areas and the subsequent quantification of the instabilities have become primary activities of seismic microzonation studies. In this context, our study has focused on the identification of

areas susceptible to soil liquefaction in Italy. We have analyzed ground-motion hazard maps and the associated hazard disaggregation to define areas where liquefaction triggering due to seismic activity can not be excluded and, therefore, further efforts are required to evaluate liquefaction susceptibility. The final result is a liquefaction triggering map showing areas with different triggering potential (Fig. 7). The information contained therein (particularly, the value of $\widehat{M}$) can be considered as an alternative to the results one can obtain by applying other approaches, particularly those proposed by the

SM Working Group (2008 and 2015; interested readers may also refer to the note of Technical Commission on Seismic Microzonation (2018)) for the determination of the reference magnitude for liquefaction susceptibility assessment. It is worth noting that, regardless of the approach used, soil liquefaction is a highly localized phenomenon whose occurrence is intimately related to site-specific geological and geotechnical conditions, which deserve focused investigations. Therefore, studies like the one presented here primarily serve as a basic guide to identifying sites where the possibility of triggering

liquefaction can be discounted within the land-use planning process.

Despite the reliability of our results, which was examined by analyzing the distribution of past liquefaction events over the Italian territory, they can certainly be refined as soon as site-specific studies become available. On the one hand, site-specific

hazard analyses (e.g., Cramer 2014, Barani et al., 2020, Mascandola et al., 2022) allow the refinement of the values of $a_{max}$. On the other hand, in-situ measurements provide a more accurate definition of the ground type for the site of interest. At

least as far as this issue is concerned, practitioners and land-use planners can interrogate our results through the web service at www.distav.unige.it/rsni/milq.php and refine of the level of liquefaction triggering potential of the site of interest by changing the ground type and the topographic class through a user-friendly interface. The service returns as output the updated values of $a_{max}$ and $\widehat{M}$, thus allowing the refinement of the triggering potential level.

**Data availability**

The results presented in this manuscript are stored in a "PostgreSQL" database and made available through the online web service at www.distav.unige.it/rsni/milq.php. They can also be provided to the Editorial Board Members, Referees, and readers upon request.

**Author contribution**

S.B. carried out most of the analyses (i.e, computation of $a_{max}$ and hazard disaggregation), wrote the main body of the
manuscript, and prepared Fig. 4 and Fig. 9. G.F. contributed to the analyses (particularly, to the computation of $a_{max}$), carried out an independent check of the values of $\widehat{M}$, and prepared all the figures (except for Fig. 4 and Fig. 9) via the QGIS software. D.S. developed the web service (www.distav.unige.it/rsni/milq.php) for the dissemination of the results. All authors reviewed the manuscript and contributed to the interpretation of the results.

**Competing interests**

The authors declare that they have no conflict of interest.

**Acknowledgments**

The regional Government of the Tuscany Liguria region funded a significant proportion of the present study. We are thankful to I. Iervolino and E. Chioccarelli for providing us with the $V_{S,Eq}$ map and the fruitful discussions.



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




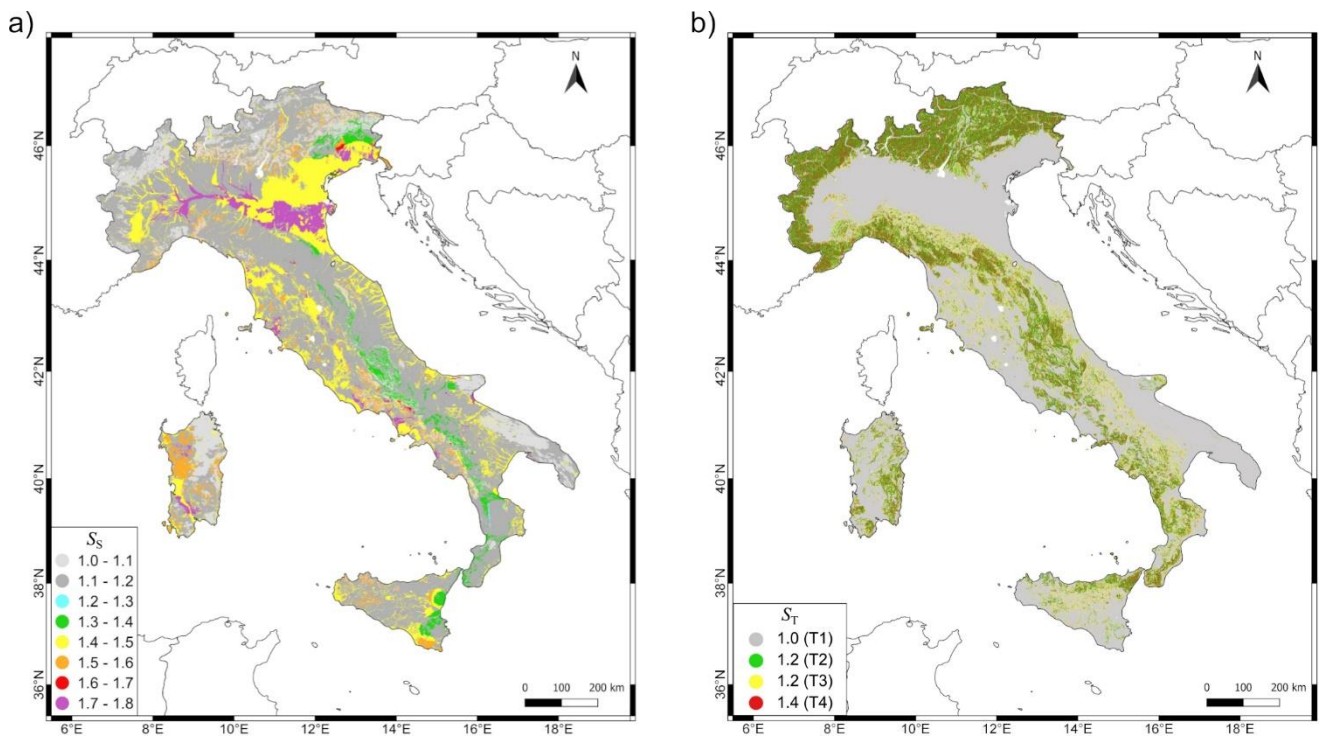

**Figure 1: (a) Soil amplification factor ($S_S$) map; (b) topographic amplification factor ($S_T$) map (topographic classes, T1 – T4, are**
**indicated in the legend).**



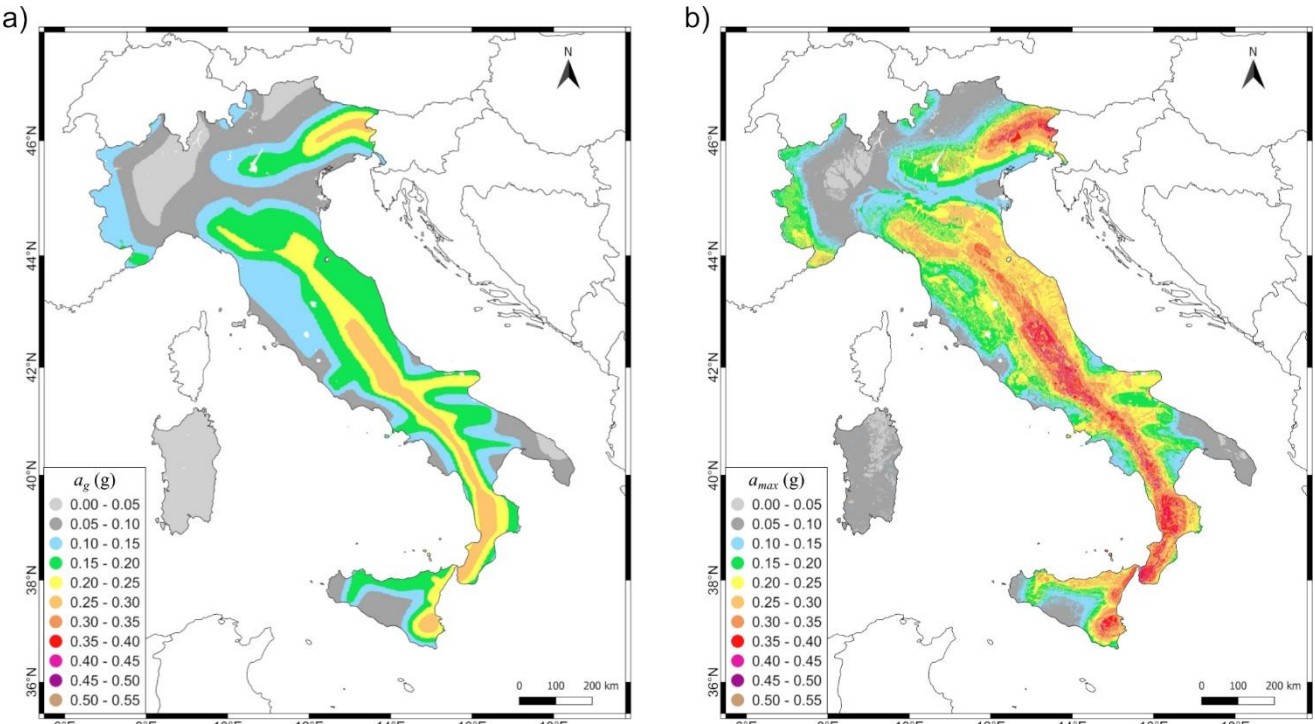

**Figure 2: (a) Peak horizontal acceleration hazard map for rock conditions and flat topography corresponding to a return period of 475 years ($a_g$ hazard map); (b) peak horizontal acceleration hazard map at the ground surface for a return period of 475 years ($a_{max}$ hazard map).**



**Figure 3: Maps of mean (left-hand column) and modal (right-hand column) magnitude values obtained from the 1D disaggregation of the 475-year spectral acceleration hazard corresponding to a response period ($T$) of 0.01s, 0.2s, and 1s.**





**Figure 4: Example probability mass functions (PMFs) of magnitude: (a) unimodal PMF with no skew; (b) negatively skewed PMF; (c) positively skewed PMF; (d) bimodal PMF with the second mode contributing to the hazard more than the mean scenario; (e) bimodal PMF with the second mode contributing to the hazard less than the mean magnitude. Mean ($\overline{M}$), modal ($M^*$), and preferred magnitude ($\widehat{M}$) are displayed in each panel. In the case of bimodal distributions (panels (d) and (e)), the first and second mode are indicated as $M_1^*$ and $M_2^*$, respectively.**





**Figure 5: Maps of preferred magnitude $\widehat{M}(T)$ (left-hand column) and related contribution to the hazard (right-hand column) for spectral periods of 0.01s, 0.2s, and 1s. The contribution to the hazard for each value of $\widehat{M}(T)$ is expressed by the probability mass function (PMF) of magnitude in that point (i.e., magnitude bin).**




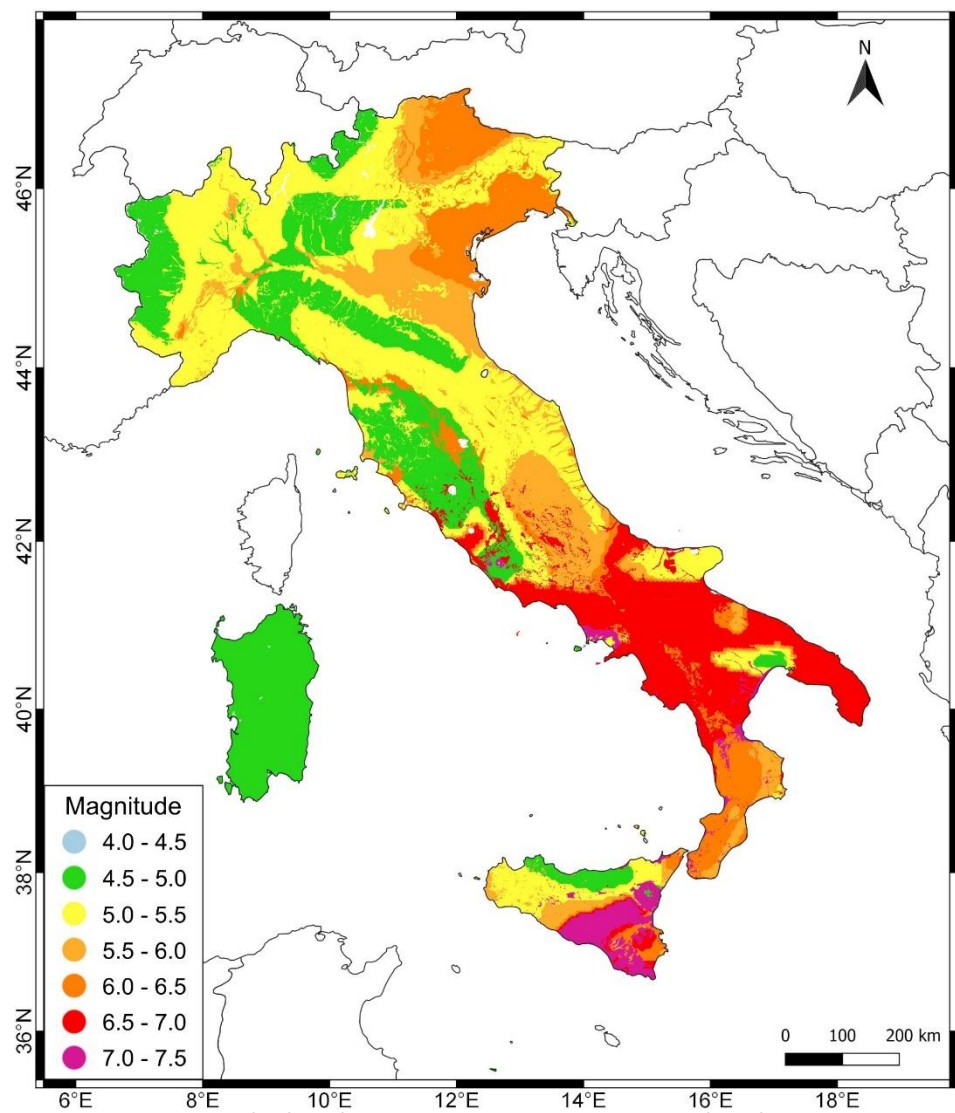

**Figure 6: Maps of preferred magnitude $\widehat{M}$: $\widehat{M} = \widehat{M}(T = 0.01s)$ for ground type A; $\widehat{M} = \widehat{M}(T = 0.2s)$ for ground type B; $\widehat{M} = \widehat{M}(T = 1.0s)$ for ground type C, D, and E (the reader may refer to the main body of the article for an explanation of the ground types). Note that the value of $\widehat{M}$ assigned to the entire Sardinia Island, for which hazard and hazard disaggregation are not available (MPS Working Group, 2004), is based on historical seismicity (characterized by rare events with magnitude less than 5 (Rovida et al., 2020 and 2022)).**



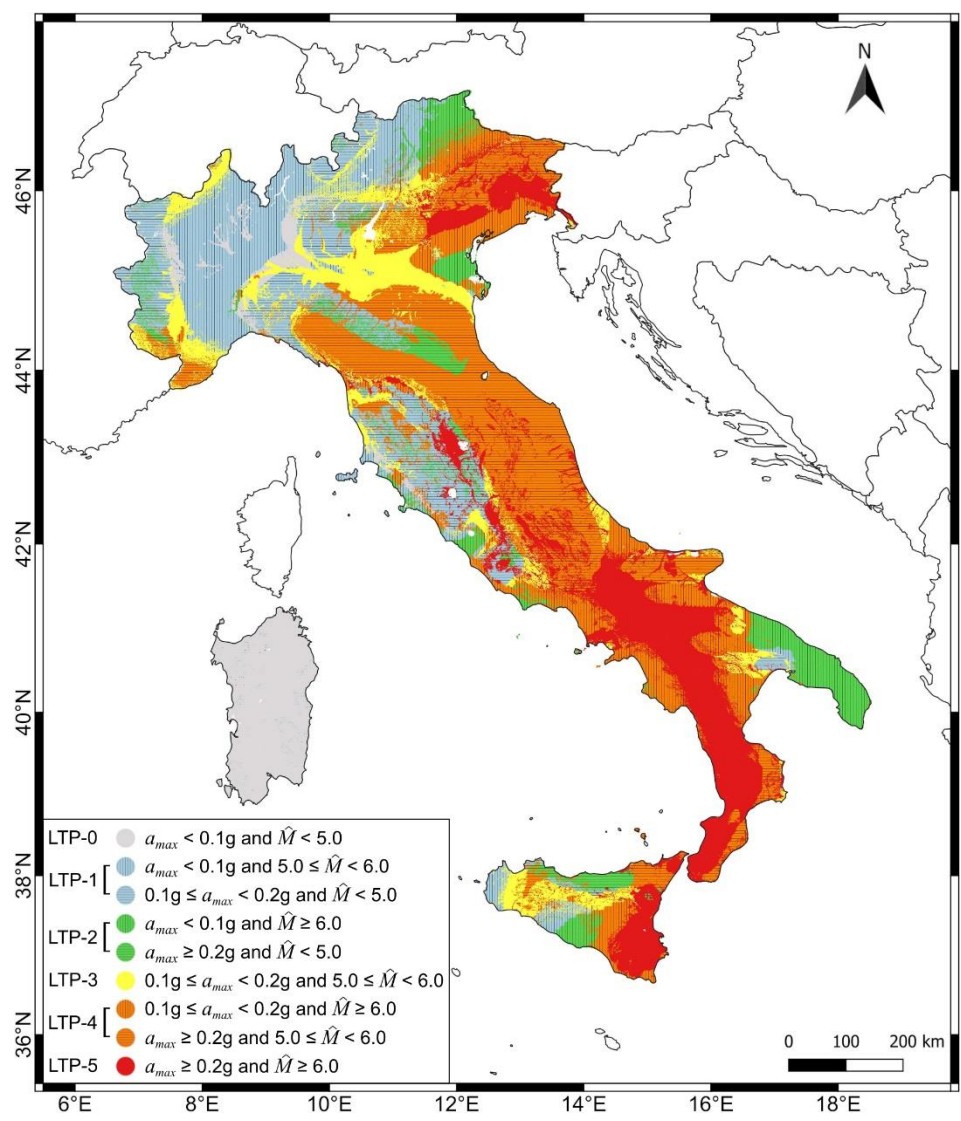

**Figure 7: Liquefaction triggering map of Italy (MILQ) for a return period of 475 years. Liquefaction triggering potential (LTP)**
**classes are indicated in the legend: no potential (LTP-0), very low potential (LTP-1), low potential (LTP-2), moderate potential**
**(LTP-3), high potential (LTP-4), and very high potential (LTP-5).**



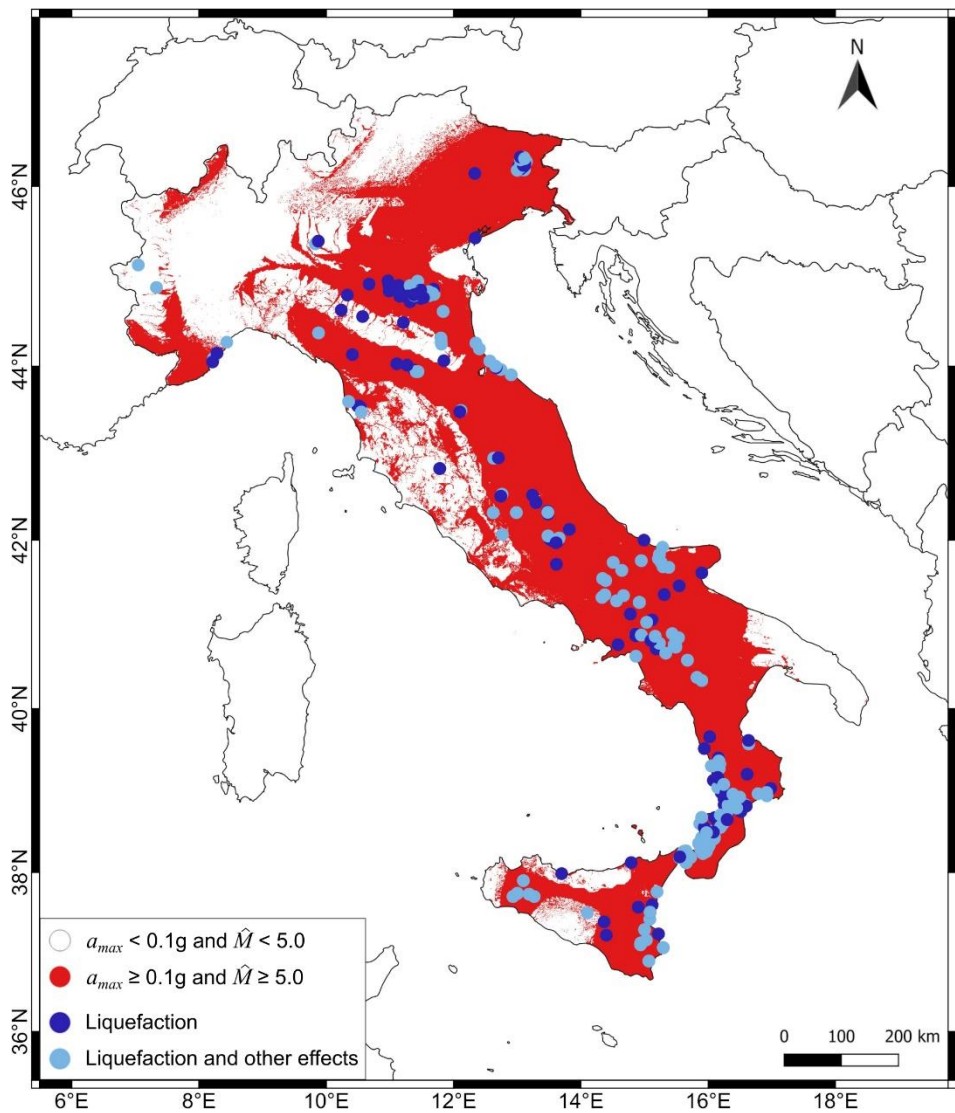

**Figure 8: Same as Fig. 7 but in red and white. Red is used when both the value of surface peak horizontal acceleration for a return period of 475 years, $a_{max}$, and the preferred magnitude, $\hat{M}$, are equal to or greater than the thresholds of 0.1g and 5.0, respectively. White is used otherwise. The historical liquefaction phenomena occurred in Italy from 1117 A.D. to 2018 (from the Italian Catalogue of Earthquake-Induced Ground Failures – CEDIT (Caprari et al., 2018)) are superimposed. The term "other effects" in the legend refers to other earthquake-induced phenomena, such as landslides, ground cracks, and surface faulting.**




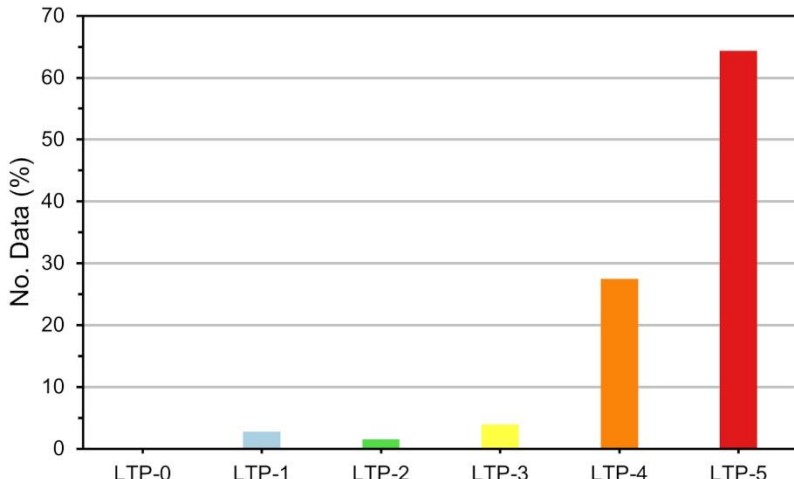


**Figure 9: Distribution of liquefaction events in Italy (from the Italian Catalogue of Earthquake-Induced Ground Failures – CEDIT (Caprari et al., 2018)) for each class of liquefaction triggering potential (LTP).**



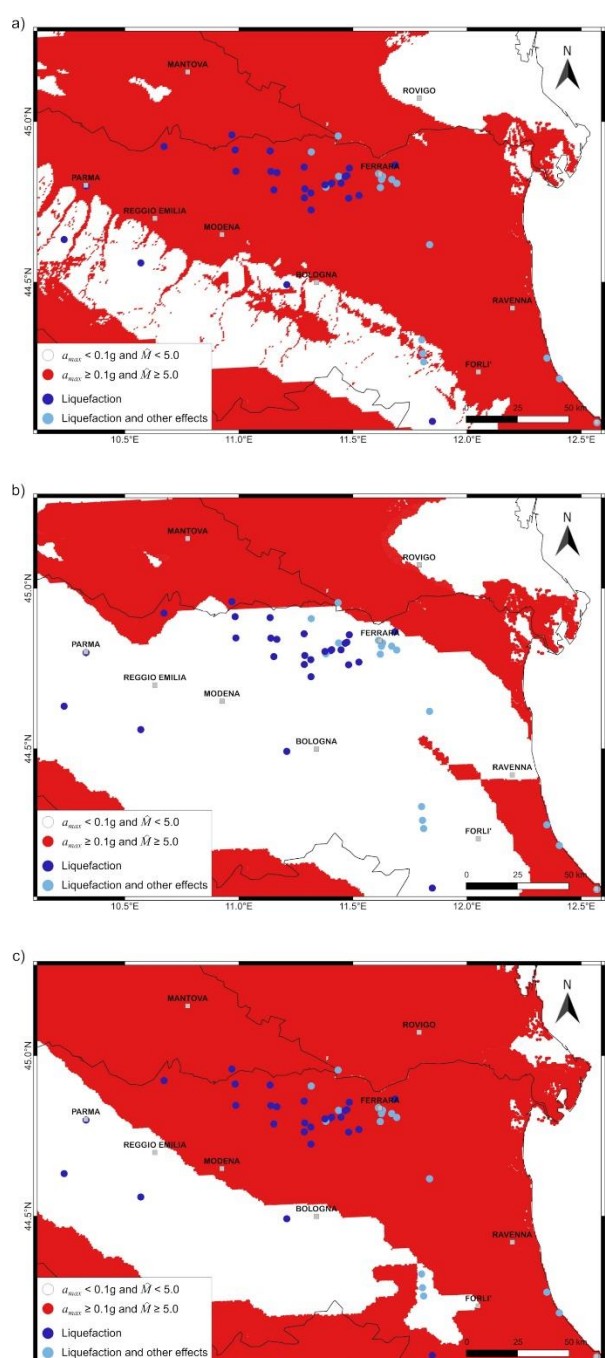


**Figure 10: Liquefaction triggering and historical liquefaction phenomena in the Po Plain area: (a) zoom of Fig. 8; (b) same as panel (a) but obtained by assuming $\widehat{M} = \widehat{M}(T = 0.01s)$ regardless of the ground type (see the map in top left corner of Fig. 5); (c) same as panel (b) but for return period of 2475 years (i.e., the values of $a_{max}$ and $\widehat{M} = \widehat{M}(T = 0.01s)$ used in input refer to a return period of 475 years).**