# Peer review of "Evaluation of liquefaction triggering potential in Italy: a seismic hazard-based approach"

_EGUsphere, 2022_

## Author Comment (AC1)

**Reviewer #1:**

General comment: the paper deals with an interesting topic (a liquefaction triggering map for Italy) and is generally well written. I think that the paper is important from a technical point of view. Further comments are reported below:

- It could be useful to define liquefaction induced by earthquakes.
  As suggested, we have added the definition of "liquefaction" at the beginning of the Introduction.

- I suggest adding further information regarding liquefaction phenomena in Italy.
  On line 53, we have added the following sentence: "According to the Italian Catalogue of Earthquake-Induced Ground Failures – CEDIT (Martino et al., 2014; Caprari et al., 2018), more than 300 liquefaction events have occurred in Italy since 1000 A. D., the last of which are associated with the 2012 Emilia seismic sequence (e.g., Scognamiglio et al., 2012; Minarelli et al. 2022). Therefore, it is of paramount importance to define a map that classifies sites as a function of their liquefaction triggering potential".

- Line 63. "Lime" should be replaced by "silt".
  As suggested, "lime" has been changed to "silt"

- Line 99. SS should be defined as the "soil stratigraphy amplification factor". "Stratigraphy" should be added.
  Following the suggestion, "soil amplification factor" has been changed to "stratigraphic amplification factor", which is the terminology used by the Italian building code.

- Line 107. I think that Fig. 1 is interesting and should be better commented. For example, further considerations could be done on the fact that the highest values of SS are in Emilia Romagna where liquefaction occurred in 2012.
- As suggested, on line 119, we have added the following sentence: "It is worth noting the highest values of Ss are concentrated in the Po Plain sedimentary basin where, as mentioned previously, several liquefaction phenomena were triggered during the 2012 Emilia seismic sequence (we refer the reader to the Discussion section for details on this area)".

- Line 116. M must be defined too.
  $M$ is defined on line 76 as "moment magnitude".

- Line 135. Maybe "di" should be replaced by "of".
  The sentence "larger magnitudes control di hazard in areas" has been changed to "larger magnitudes control the hazard in areas".

---

## Author Comment (AC2)

**Reviewer #2:**

The proposed manuscript presents a map of Italy where regions with different seismic intensity levels able to trigger (or not) earthquake-induced liquefaction are distinguished for a return period of 475 years. The study has a good scientific significance, where the substantial contribution and main novelty can be detected in the innovative criteria to select the moment magnitude to use for assessing the soil liquefaction potential according to the simplified approach.

The technical approach and the applied methods are valid, i.e., the scientific quality is good. The presentation quality is generally fair, but the technical and English language needs to be improved as detailed in the 'technical corrections'.

Even though the paper addresses relevant questions within the scope of NHESS, I raised some critical issues that require major revisions before publication, as listed in the following:

SPECIFIC COMMENTS

- The Authors correctly said that if the seismic hazard at a specific site is too low, the liquefaction does not need to be considered in the planning process. Conversely, the opposite situation (i.e., high seismic hazard) is necessary but not enough to trigger liquefaction phenomena. Indeed, earthquake-induced liquefaction occurs only if the triggering (earthquakes) and predisposing factors (loose saturated granular soils) coexist contemporarily. For the above reason, I think that the title of the paper is misleading because it seems that the proposed map shows the areas where there is the occurrence of liquefaction. This is not true because liquefaction is going to happen only where susceptible soils to liquefaction are encountered. Consequently, I suggest a much more robust description of the title and the map, such as an example: 'Screening map for earthquake-induced liquefaction based on the existence of triggering factors' that in Italian would be 'Mappa di selezione delle aree in cui sussistono fattori scatenanti della liquefazione sismo-indotta'. Please, revise the manuscript accordingly.

  As is correctly observed, liquefaction occurs if both triggering and predisposing geological conditions coexist. The term "triggering" (or "triggering factors") is commonly used to indicate the existence of certain seismic conditions that, whether met, can induce liquefaction in cohesionless, saturated soils. However, cohesion and saturation are not triggering factors but predisposing geological conditions that define the susceptibility of the material to liquefy. Hence, the use of the word "triggering" in the abstract should avoid misunderstanding. However, we agree with the reviewer that the original title is not fully informative but the one proposed, in our opinion, is convoluted and do not reflect completely the scope of work. Therefore, we have changed the original title to: "Evaluation of liquefaction triggering potential in Italy: a seismic hazard-based approach".

- The Authors correctly stated in the conclusions that site-specific analyses can allow the refinement of the maximum acceleration, and that subsoil and topographic class can be changed through a user-friendly interface. However, it can be appropriate to highlight that a seismic response analysis to assess the local effects is mandatory when the considered site cannot be categorized in one of the subsoil classes according to the national building code (section 3.2.2, Ministero delle Infrastrutture e dei Trasporti, 2018).

  We thank the reviewer for this important observation. Indeed, in addition to the Italian norm, also the ICMS document requires the use of site-specific amplification factors, either determined using the so-called abacuses (pre-compiled tables providing the user with amplification factor values for different soil characteristics) or specific ground-response analysis. Therefore, we have modified the MILQ web service, so as to allow the user to enter a site-specific amplification factor value. The manuscript has been modified accordingly (end of section 5 and end of the conclusions).

- Putting the acronym in the title is probably not the best solution because it is not informative. It can be explained in the subsequent part of the paper.
  Following the first comment (see our reply), we have changed the title.

- The Italian acronym is described in Italian, it is suggested to also report the English translation to be much clearer and to be in line with international standards.
  Done throughout the manuscript.

- The Author said in the abstract that they refer to the triggering criteria proposed by the Italian Guidelines for Seismic Microzonation, but nothing is said about the predisposing factors (type of soil, shallow ground water table). I suggest that the Authors immediately clarify in the Abstract if the predisposing factors are considered or not in the definition of the map.
  Following the suggestion, in the very beginning of the abstract we have specified the following: "To this end, we refer to the triggering criteria (not to be confused with liquefaction susceptibility criteria, which essentially take into account soil type and depth to groundwater) proposed by the Italian Guidelines for Seismic Microzonation".

TECHNICAL CORRECTIONS

- First line of the Introduction. 'potential liquefaction hazards' is redundant. I suggest 'potential liquefaction' or 'liquefaction hazard' which are both standalone.
  Following the suggestion, "potential liquefaction hazards" has been changed to "liquefaction hazards".

- First line of the Introduction. 'site' is repeated two times in both (1) and (2) questions. I suggest being concise and anticipating 'at the site of interest' before 'always'. In this way, the two questions are becoming brief and more effective without 'at the site of interest' in question (1) and 'at that site' in question (2), respectively.
  We have modified the sentence according to the suggestion above.

- Second line of the Introduction. 'favorable' could be better substituted with 'prone'.
  According to the suggestion, "favorable" has been changed to "prone".

- Second line of the Introduction. 'ground motions' is redundant and can be removed. 'Earthquakes' should be instead of 'earthquake'.
  We have modified the sentence according to the suggestion above.

- Line 9 of the Introduction (section 25 of the pdf file). A bracket is open after 'surface' but it is not closed anymore. Please, correct this.
  The bracket was a typo. The sentence is "peak horizontal acceleration at the ground surface for the return period of interest…"

- First line of section 30 of the pdf file. 'site hazard' should be 'seismic hazard' otherwise it is unknown what kind of hazard the Authors refer to.
  As suggested, "site hazard" has been changed to "seismic hazard".

- Line 4 of section 45 of the pdf file. What do the Authors mean by '1D magnitude'?
  "1D" has been deleted. As specified later in the manuscript, magnitude contributions result from 1D disaggregation (i.e., magnitude disaggregation).

- Line 2 of section 50 of the pdf file. The year of the reference 'Italian Guidelines for Seismic Microzonation' needs to be added.

On line 50, we have specified that the Italian Guidelines for Seismic Microzonation were originally released by the SM Working Group in 2008. The 2015 version is just the English edition.

- Chapter 2. Criteria for liquefaction triggering. Line 3. 'limes' and 'limey' should be substituted with 'silts' and 'silty', respectively.
  As suggested "limes" and "limey" have been changed to "silts" and "silty", respectively.

- Chapter 3.1 Seismic hazard map. Line 5 of section 100 of the pdf file. 'ground types' should be 'soil types' or 'subsoil class'. Please, revise is in the entire manuscript.
  The term "ground type" is currently used by Eurocode 8. Hence, on line 114, we have specified the following "…ground types (also referred to as "soil types" or "subsoil classes").

- Chapter 3.2 Hazard disaggregation. Line before section 130 of the pdf file. The statement 'We expect that decreasing $V_{S,eq}$ and/or increasing soil thickness the site resonance period increases.' is due to the definition of the natural period of a homogenous deformable soil deposit overlying a half-space. Please, add the reference for this, e.g., Kramer (1996).
  As suggested, we have added the reference to Kramer (1996).

- Line section 135 of the pdf file. 'di' after control needs to be deleted.
  The sentence "larger magnitudes control di hazard in areas" has been changed to "larger magnitudes control the hazard in areas". "di" was a typo.

- Chapter 6. Discussion. Line 2 of section 220 of the pdf file. 'with red dots indicating the sites where liquefaction triggering is expected to occur' the dots looks blue and light blue and are used to define past liquefaction events. Please, revise.
  The sentence is revised as follows: "These events have been superimposed on that map (blue and light blue dots), which is now displayed in Fig. 8 using a more intuitive red and white scale, with the red color indicating the sites where liquefaction triggering is expected to occur".

MENTIONED AND SUGGESTED REFERENCES

Ministero delle Infrastrutture e dei Trasporti. Aggiornamento delle "Norme tecniche per le costruzioni", D.M. 17 Gennaio 2018, Supplemento ordinario alla Gazzetta Ufficiale N. 42 del 20 Febbraio, 2018.

Kramer S.L. 'Geotechnical Earthquake Engineering' Prentice-Hall, Upper Saddle River, NJ 1996.

While the first suggested reference was already mentioned in the manuscript, we have added the one of Kramer.